# Negative regulation of conserved *RSL* class I bHLH transcription factors evolved independently among land plants

**Suvi Honkanen[1,2], Anna Thamm[1], Mario A Arteaga-Vazquez[3], Liam Dolan[1]\***

[1]Department of Plant Sciences, University of Oxford, Oxford, United Kingdom; [2]Australian Research Council Centre of Excellence in Plant Energy Biology, University of Western Australia, Perth, Australia; [3]Laboratory of Epigenetics and Developmental Biology, Instituto de Biotecnología y Ecología Aplicada, Universidad Veracruzana, Colonia Emiliano Zapata, Mexico

**Abstract** Basic helix-loop-helix transcription factors encoded by *RSL* class I genes control a gene regulatory network that positively regulates the development of filamentous rooting cells – root hairs and rhizoids – in land plants. The GLABRA2 transcription factor negatively regulates these genes in the angiosperm *Arabidopsis thaliana*. To find negative regulators of *RSL* class I genes in early diverging land plants we conducted a mutant screen in the liverwort *Marchantia polymorpha*. This identified FEW RHIZOIDS1 (MpFRH1) microRNA (miRNA) that negatively regulates the *RSL* class I gene Mp*RSL1*. The miRNA and its mRNA target constitute a feedback mechanism that controls epidermal cell differentiation. MpFRH1 miRNA target sites are conserved among liverwort *RSL* class I mRNAs but are not present in *RSL* class I mRNAs of other land plants. These findings indicate that while *RSL* class I genes are ancient and conserved, independent negative regulatory mechanisms evolved in different lineages during land plant evolution.

DOI: https://doi.org/10.7554/eLife.38529.001

**\*For correspondence:**
liam.dolan@plants.ox.ac.uk

**Competing interests:** The authors declare that no competing interests exist.

## Introduction

Gene regulatory networks control differentiation of cells and tissues during development. Consequently, changes in gene regulatory network structure are correlated with changes in morphology during the course of evolution. Changes in negative regulation of a gene regulatory network are a potential source of evolutionary novelty that may be associated with the evolution of new structures, or the co-option of pre-existing regulatory pathways to control novel functions. The evolution of specialised rooting structures were key morphological innovations that occurred among the first plants when they colonised the land sometime more than 470 Million years ago (*Kenrick and Crane, 1997*; *Morris et al., 2018*). The rooting structures of the first land plants are likely to have comprised systems of tip-growing filamentous cells called rhizoids that are morphologically similar to root hairs of vascular plants (*Edwards et al., 1995*; *Wellman et al., 2003*; *Taylor et al., 2005*; *Kerp et al., 2003*; *Taylor, 1995*).

The genetic program for the development of root hair cells and rhizoid cells is activated by functionally conserved basic helix-loop-helix (bHLH) transcription factors encoded by *RSL* class I genes. *RSL* class I genes promote root hair differentiation in the vascular plants rice (*Oryza sativa*), *Brachypodium distachyon* and *Arabidopsis thaliana* (*Zalewski et al., 2013*; *Kim et al., 2017*; *Menand et al., 2007*). Similarly, *RSL* class I genes positively regulate rhizoid precursor cell differentiation and rhizoid development in the liverwort *Marchantia polymorpha* and the moss *Physcomitrella patens* (*Proust et al., 2016*; *Menand et al., 2007*). Root hairs and rhizoids initiate from those epidermal cells that express *RSL* class I genes (*Zalewski et al., 2013*; *Kim et al., 2017*; *Menand et al.,*

**eLife digest** Plants colonised the land sometime more than 500 million years ago. The ancestors of the first land plants were algae that were most likely simple with a few different types of cell. Yet, when faced with the challenges of life on land, plants evolved new cell types and specialised structures with roles such as anchorage, nutrient uptake and gas exchange.

Many of these specialised structures, including the root hairs and rhizoids that allow plants to collect water and minerals from the soil, first develop as outgrowths from cells in the outer layer of the plant. An ancient and conserved mechanism activates the development of these outgrowths via genes belonging to a group known as *RSL* class I.

In the flowering plant *Arabidopsis thaliana*, a protein switches off *RSL* class I genes in a subset of these outer cells, to stop too many root hairs forming. To see whether this kind of negative regulation is also conserved among land plants, Honkanen et al. looked for regulators of *RSL* class I genes in liverworts. Small and without flowers, liverworts are a group of plants that first appeared during the earliest stages of land plant evolution.

Honkanen et al. discovered that *RSL* class I genes in liverworts are negatively regulated by a molecule named FEW RHIZOIDS1 (or FRH1). However, rather than being a protein, FRH1 is a microRNA – a short strand of genetic code that reduces how much protein is produced from a given gene. The FRH1 microRNA is conserved among liverworts and most likely evolved very early in the history of these plants. The findings indicate that different groups of land plants have evolved different negative regulators to control the conserved genes behind some of the specialised structures crucial to life on land.

DOI: https://doi.org/10.7554/eLife.38529.002

*2007*; *Jang et al., 2011*). Furthermore, constitutive over-expression of *RSL* class I genes is sufficient to modulate root hair patterning by inducing root hair development from any root epidermal cell in the monocots *O. sativa* and *B. distachyon* (*Zalewski et al., 2013*; *Kim et al., 2017*). Similarly, constitutive over-expression of *RSL* class I genes in the liverwort *M. polymorpha* and moss *P. patens* is sufficient to induce rhizoid formation from almost any epidermal cell of the gametophyte (*Proust et al., 2016*; *Jang et al., 2011*). These findings indicate that *RSL* class I genes function as molecular switches that are both necessary and sufficient to induce the root hair cell and rhizoid cell developmental program in epidermal cells.

The demonstration that the *RSL* class I genes are required for rhizoid and root hair development indicates that their function in regulating the development of filamentous rooting cells is conserved among land plants. This suggests that the *RSL* class I mechanism is likely to be ancient and have originated in the common ancestor of all land plants. *RSL* class I genes from the moss *P. patens*, liverwort *M. polymorpha* or monocot rice restore root hair development in the roothairless *A. thaliana* mutants that lack *RSL* class I gene function (*Menand et al., 2007*; *Proust et al., 2016*; *Kim et al., 2017*). This indicates that the molecular function of *RSL* class I proteins is conserved among these lineages. However, it is not known if the factors that regulate *RSL* class I gene expression are also conserved.

Negative regulation of *RSL* class I genes in *A. thaliana* suppresses the formation of root hairs in a subset of root epidermal cells (*Lin et al., 2015*). The *A. thaliana* root epidermis comprises two cell types; trichoblasts that go on to differentiate into root hair cells and atrichoblasts that differentiate as root hairless epidermal cells. *RSL* class I genes, At*RHD6* and At*RSL1*, are expressed early in development of trichoblasts before root hairs emerge and they positively regulate the expression of genes involved in root hair differentiation (*Menand et al., 2007*; *Yi et al., 2010*). At*RHD6* and At*RSL1* expression is repressed in atrichoblasts by the homeodomain protein GLABRA2 (GL2) (*Di Cristina et al., 1996*; *Bernhardt et al., 2005*; *Bernhardt et al., 2003*; *Koshino-Kimura et al., 2005*; *Lin et al., 2015*). At*GL2* proteins directly bind to L1 box motifs on the promoters of At*RHD6* and At*RSL1* and inhibit root hair initiation by repressing the transcription of these positive regulators of root hair development (*Lin et al., 2015*). Therefore, negative regulation has a key role in defining where *RSL* class I genes are expressed during the establishment of cell differentiation patterns in the *A. thaliana* root epidermis. However, nothing is known about the negative regulation of *RSL* class I

genes in any other land plant. The role of *GL2* as negative regulator of *RSL* class I genes and root hair development is not likely to be widely conserved because the closest homologs of *GL2* in many other vascular plants are not expressed in roots (*Huang et al., 2017*). Furthermore, *GL2* genes have not been identified in bryophytes (liverworts, mosses and hornworts) (*Zalewski et al., 2013*). Therefore, it is unclear how *RSL* class I gene expression is controlled in these lineages.

To determine if the same mechanism repressed *RSL1* class I expression in liverworts and angiosperms we took a forward genetic approach to identify negative regulators of rhizoid development in the liverwort *M. polymorpha*. We had previously screened for T-DNA mutants that developed ectopic rhizoids resulting from loss-of-function mutations in negative regulators of Mp*RSL1*. However, all ectopic rhizoid mutants identified harboured gain-of-function mutations in the positive regulator of rhizoid development, Mp*RSL1* (*Proust et al., 2016*), indicating that this approach was unlikely to identify loss-of-function mutations in negative regulators. Therefore, we opted to identify negative regulators of Mp*RSL1* expression by screening for gain-of-function mutations in the genes encoding these repressors. We screened for rhizoidless mutants resulting from overexpression of negative regulators and identified four gain-of-function mutations in a gene encoding a miRNA. Here we describe the discovery of FEW RHIZOIDS1 (FRH1), a novel microRNA (miRNA) that negatively regulates the liverwort *M. polymorpha* single copy *RSL* class I gene Mp*RSL1*. Our results demonstrate that a lineage specific mechanism mediated by the MpFRH1 miRNA controls the expression of a functionally conserved *RSL* class I transcription factor Mp*RSL1* in liverworts. These results suggest that while the role of *RSL* class I genes as positive regulators of filamentous rooting cell development is conserved, distinct mechanisms that repress *RSL* class I expression have evolved among different lineages of land plants.

## Results

### *FEW RHIZOIDS1* is a novel regulator of rhizoid precursor cell differentiation in *M. polymorpha*

Mp*RSL1* is a master regulator of the development of structures that originate from single epidermal cells – rhizoids, mucilage papillae and gemmae – in the liverwort *M. polymorpha* (*Proust et al., 2016*). To identify regulators of Mp*RSL1*, we screened a population of 150,000 of *M. polymorpha* T-DNA insertion lines for rhizoidless phenotypes. We identified four mutants, ST21-1, ST33-2, ST45-2 and ST49-10 that develop no or very few rhizoid precursor cells and consequently develop few rhizoids (*Figure 1A–L*). Two mutants (ST45-2 and ST49-10) were backcrossed to wild type and the phenotypes of individuals from the F1 generation were scored. The few rhizoids phenotype segregated as a single locus in each F1 generation, that is 50% of plants developed few rhizoids and 50% were indistinguishable from wild type (*Figure 1—figure supplement 1*). Furthermore, the few rhizoids phenotype co-segregated with the hygromycin resistance marker gene on the T-DNA in both ST45-2 and ST49-10 segregating F1 populations (*Figure 1—figure supplement 1*). Based on the ratios of hygromycin resistant to hygromycin sensitive F1 plants, there is a single T-DNA insertion that is tightly linked to the few rhizoids phenotype in ST45-2. There are two T-DNA insertions in ST49-10; one responsible for the few rhizoids phenotype and a second unlinked T-DNA insertion.

To identify the T-DNA insertion sites in the few rhizoids mutants, we isolated the T-DNA insertion flanking genomic sequences by thermal asymmetric interlaced PCR (TAIL PCR). We identified a T-DNA insertion within a 1.2 kb region of the *M. polymorpha* genome in each of the few rhizoids mutants ST21-1, ST33-2, ST45-2 and ST49-10 (*Figure 1M*, *Supplementary file 1*), suggesting that this genomic region is involved in rhizoid precursor cell differentiation. Mapping the *M. polymorpha* gametophyte transcriptome sequences onto the genome assembly (*Honkanen et al., 2016*) indicated that no transcript overlapped with the T-DNA insertion sites in any of the four few rhizoids mutants. Therefore, we hypothesised that the T-DNA insertions in these mutants may have altered the expression level of a nearby transcript. To test this hypothesis, we used qRT-PCR analysis to measure the steady state transcript levels of all genes transcribed within the 15 kb genomic interval around the T-DNA insertion sites in the few rhizoids mutants. A transcript fragment that mapped next to the right border of the T-DNA insertions was expressed at higher levels in each of the few rhizoids mutants ST21-1, ST33-2, ST45-2 and ST49-10 than in wild type (*Figure 1N*). To identify the full-length transcript we performed 3' and 5' RACE-PCR. This identified a 1.2 kb intron-less

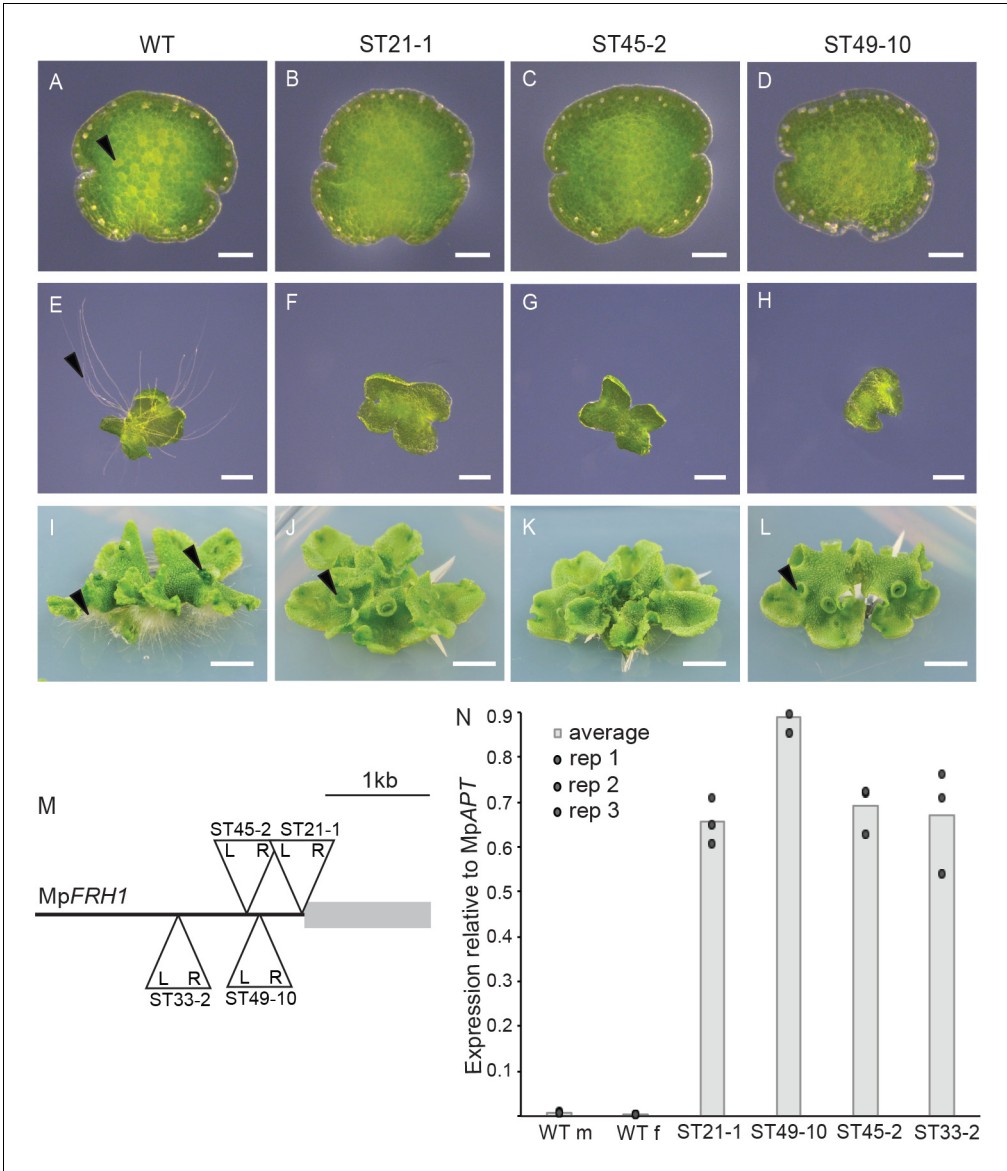

**Figure 1.** T-DNA insertion within the Mp*FRH1* promoter results in elevated steady state levels of Mp*FRH1* mRNA and defective rhizoid precursor cell differentiation. (**A–L**) Phenotype of wild type *M. polymorpha* and the few rhizoid mutants ST21-1, ST45-2 and ST49-10. (**A–D**) One day old gemma (scale bar 100 µm), (**E–H**) four day old gemma (scale bar 1 mm), (**I–L**) 28 day old gemma (scale bar 5 mm) of wild type (**A, E, I**) and the few rhizoid mutants ST21-1 (**N, F, J**), ST49-10 (**C, G, K**) and ST45-2 (**D, H, J**). The arrowheads indicate rhizoid precursor cells (in A-D) rhizoids (in E-L) and gemma cups (in I-L). (**M**) Location and orientation of the T-DNA insertion sites within the Mp*FRH1* locus. L and R stand for T-DNA left and right border, respectively. (**N**) qRT-PCR analysis of steady state Mp*FRH1* mRNA levels in 15 day old gemmae of wild type and the few rhizoid mutants ST21-1, ST45-2 and ST49-10. The *MpFRH1* transcript level was normalised against Mp*APT1*.

DOI: https://doi.org/10.7554/eLife.38529.003

The following figure supplement is available for figure 1:

**Figure supplement 1.** The few rhizoids phenotype of mutants ST45-2 and ST49-10 co-segregates with the hygromycin resistance marker on the T-DNA.

DOI: https://doi.org/10.7554/eLife.38529.004

polyadenylated transcript (*Supplementary file 2*) that we named *FEW RHIZOIDS1* (Mp*FRH1*). Together these data indicate that in each of the four few rhizoids mutants a T-DNA is inserted 5' of a gene encoding a 1.2 kb transcript that is expressed at higher levels than in wild type.

## *MpFRH1* negatively regulates the development of rhizoid precursor cells and rhizoids

We hypothesized that elevated expression of Mp*FRH1* was responsible for the few rhizoids phenotype in ST21-1, ST33-2, ST45-2 and ST49-10 because the 1.2 kb Mp*FRH1* transcript was more abundant in each of these mutants than in wild type. To test this hypothesis, we expressed the Mp*FRH1* transcript under the control of the strong constitutive rice *ACTIN1* promoter in wild type *M. polymorpha*. The majority of the resulting transformed plants (88 out of 103) developed very few or no rhizoids (*Figure 2C,G,K*). The steady state transcript levels of Mp*FRH1* were measured in two week old gemmae from eight transformed lines; four transformed lines with very few rhizoids and four transformed lines with wild type phenotype. Steady state levels of Mp*FRH1* transcript were higher than wild type in each of the transformed lines with few rhizoids, while wild type levels of Mp*FRH1* transcript were observed in the lines with wild type phenotype (*Figure 2—figure supplement 1*). This indicates that over-expression of the 1.2 kb Mp*FRH1* transcript in the wild type background was sufficient to reproduce the few rhizoids phenotype of the ST21-1, ST49-10, ST45-2 and ST33-2 T-DNA insertion mutants. This suggests that each is a gain-of-function mutant, in which elevated expression of Mp*FRH1* results in reduced number of rhizoid precursor cells and rhizoids. Therefore, the ST21-1, ST49-10, ST45-2 and ST33-2 mutants were re-named as Mp*FRH1^{GOF1}*, Mp*FRH1^{GOF2}*, Mp*FRH1^{GOF3}* and Mp*FRH1^{GOF4}*, respectively. Together these findings indicate that Mp*FRH1* is a negative regulator of rhizoid precursor cell differentiation.

## Mp*FRH1* negatively regulates the development of epidermal papillae and gemmae

The phenotype of the Mp*FRH1* gain-of-function mutants and $_{pro}$*ACT:*Mp*FRH1* plants was similar to Mp*rsl1* loss-of-function mutant phenotype (*Figure 2*). While Mp*FRH1* gain-of-function mutants and $_{pro}$*ACT:*Mp*FRH1* lines develop few rhizoids like Mp*rsl1*, they were also defective in the formation of other structures that originate from single epidermal cells. Wild type *M. polymorpha* plants develop vegetative reproductive propagules called gemmae in cup-like structures called gemma cups (*Figure 2M*). In the wild type, gemmae develop from epidermal cells at the bottom of each gemma cup (*Proust et al., 2016*). By contrast, gemmae only rarely develop in Mp*rsl1*, Mp*FRH1^{GOF}*, $_{pro}$*ACT:*Mp*FRH1* lines and the gemma cups were generally empty (*Figure 2N–P*). Furthermore, like Mp*rsl1* loss-of-function mutants, Mp*FRH1^{GOF}* mutants and $_{pro}$*ACT:*Mp*FRH1* lines lacked multicellular mucilage papillae that develop in the epidermis near meristematic regions of wild type plants (*Figure 2Q–T*). Together, these data suggest that Mp*FRH1* negatively regulates the development of the same structures – rhizoids, gemmae and epidermal papillae – that are positively regulated by Mp*RSL1*.

## The Mp*FRH1* transcript encodes a microRNA (miRNA)

There are no long open reading frames in the 1.2 kb Mp*FRH1* transcript sequence and the sequence is not similar to any Arabidopsis protein coding sequences. Therefore, we hypothesized that Mp*FRH1* might function as an RNA. Consistent with this hypothesis, the 21 nt predicted miRNA mpo-MIR11861 (UGUGUGAGAAGAGGCCAAUGU) maps to the same genomic location as the Mp*FRH1* transcript (*Tsuzuki et al., 2016*). To verify that the predicted miRNA was responsible for the few rhizoids phenotype of the Mp*FRH1^{GOF}* lines, we over-expressed a 150 bp fragment of the Mp*FRH1* transcript containing the predicted miRNA hairpin structure (*Figure 3A*) in the wild type background. Over-expression of the 150 bp miRNA hairpin-containing fragment (Mp*FRH1^{miRNA}*) in the wild type background was sufficient to reproduce the few rhizoids phenotype (*Figure 4C,I,O*, *Figure 4—figure supplement 1*), suggesting that Mp*FRH1* is a miRNA gene. The presence of the mature MpFRH1 miRNA was verified using stem-loop PCR (*Figure 3—figure supplement 1*). In the stem-loop PCR a 60 bp band corresponding to the MpFRH1 miRNA and a fused stem-loop sequence was stronger in the Mp*FRH1* gain-of-function mutant samples compared to wild type, suggesting that elevated levels of the Mp*FRH1* pri-miRNA transcript in these mutants give rise to

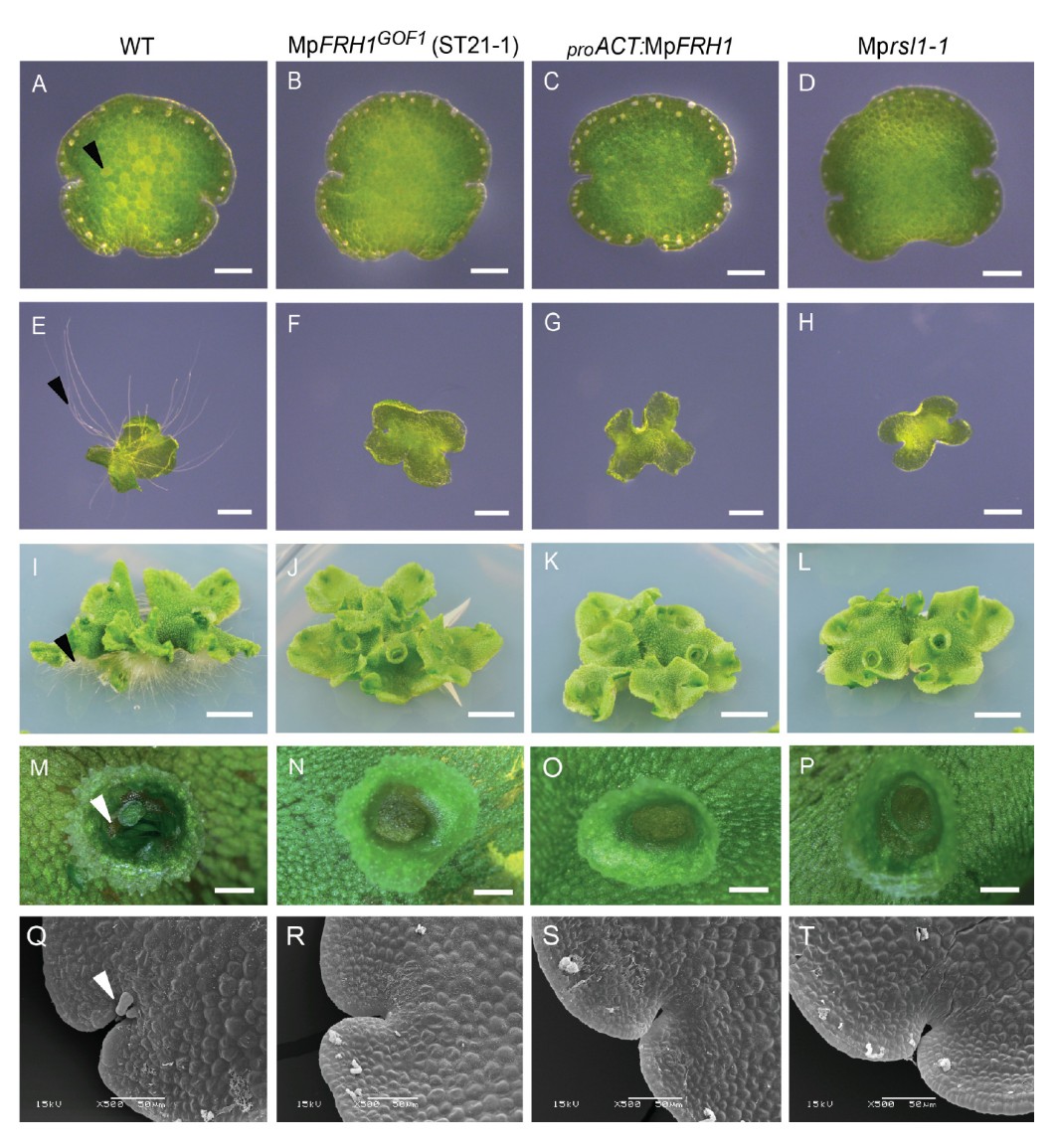

**Figure 2.** The phenotype of the Mp*FRH1* gain-of-function mutants and plants transformed with *proACT*:Mp*FRH1* is similar to Mp*rsl1* loss-of-function mutant phenotype. (**A–T**) Phenotype of wild type *M. polymorpha*, T-DNA insertion line Mp*FRH1*^*GOF1*/ST21-1, plant transformed with *proACT*:Mp*FRH1* and Mp*rsl1-1* loss-of-function mutant. One day old gemma (A-D, scale bar 100 μm), four day old gemma (E-H, scale bar 1 mm), 28 day old gemma (I-L, scale bar 5 mm), gemma cup of mature plant (M-P, scale bar 600 μm) and meristematic region of one day old gemma (Q-T, scale bar 50 μm) of wild type (**A, E, I, M, Q**), Mp*FRH1*^*GOF1*/ST21-1, (**B, F, J, N, R**), *proACT*:Mp*FRH1* (**C, G, K, O, S**) and Mp*rsl1-1* (**D, H, L, P, T**). The arrowheads indicate rhizoid precursor cells (in **A-D**), rhizoids (in **E-L**), gemmae (in **M-P**) and mucilage papillae (in **Q-T**).

DOI: https://doi.org/10.7554/eLife.38529.005

The following figure supplement is available for figure 2:

**Figure supplement 1.** Over-expression of the full-length Mp*FRH1* transcript behind the strong constitutive rice *ACTIN* promoter in the wild type background is sufficient to reproduce the few rhizoids phenotype.

DOI: https://doi.org/10.7554/eLife.38529.006

elevated levels of mature MpFRH1 miRNA (*Figure 3—figure supplement 1*). There are three putative small open reading frames (PSORFs) of 47, 49 and 71 amino acids on the MpFRH1 transcript. To verify that MpFRH1 does indeed function as miRNA and does not produce a small peptide, we generated a version of MpFRH1 in which the putative start codons (ATG) for each of these PSORFs were

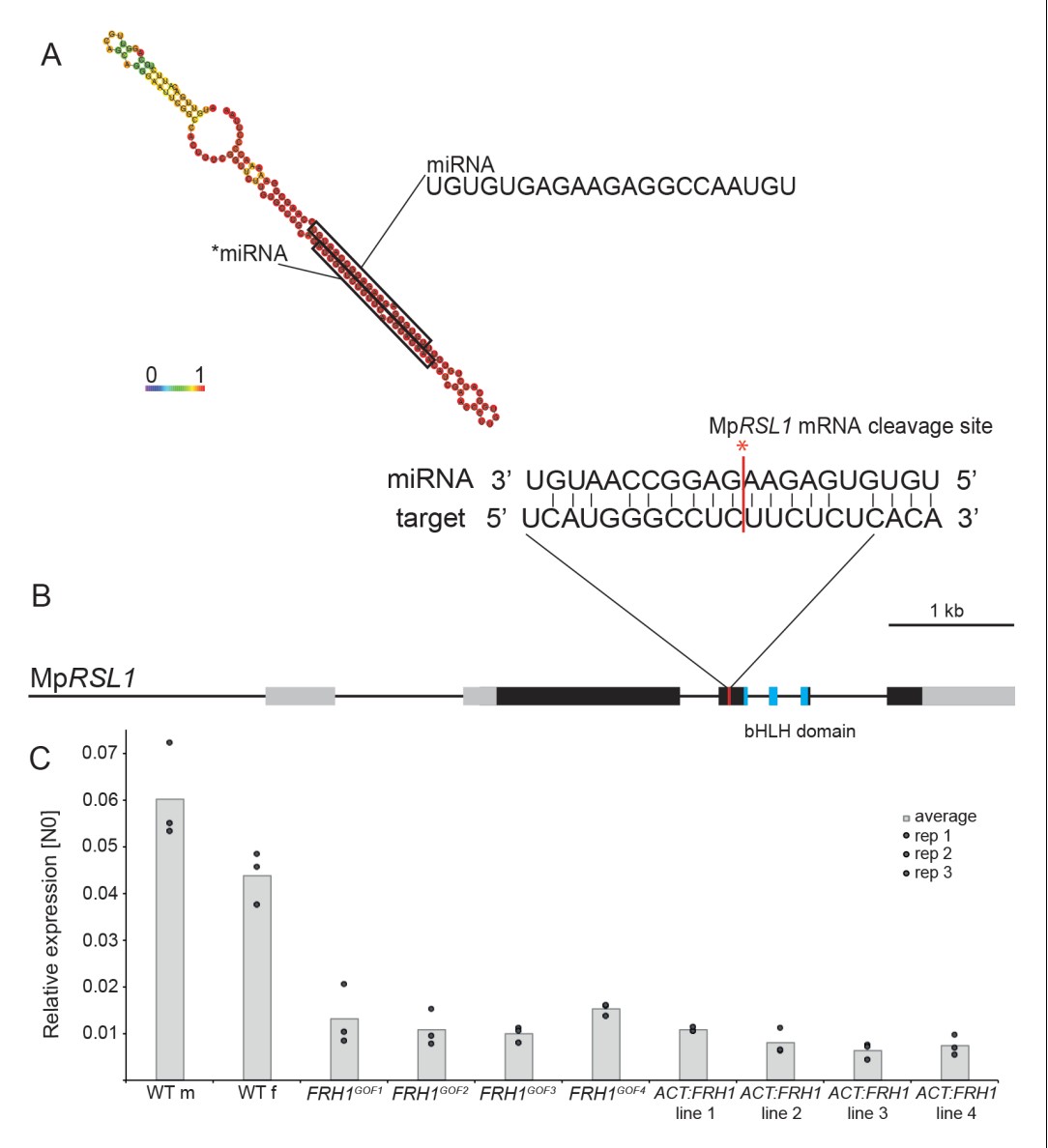

**Figure 3.** The Mp*FRH1* locus produces a miRNA that targets the Mp*RSL1* transcript. (**A**) RNA folding prediction of the 150 bp sequence sufficient to reproduce the few rhizoids phenotype when over-expressed in the wild type carried out using RNAfold (*Gruber et al., 2008*). The colours represent base-pairing probabilities. The small RNA sequences corresponding to MpFRH1 miRNA (mpo-MIR11861) and the complementary *miRNA are indicated. (**B**) Mp*RSL1* gene model, the MpFRH1 miRNA target site is indicated in red. (**C**) Mp*FRH1* negatively regulates Mp*RSL1* transcript level. qRT-PCR quantification of steady state Mp*RSL1* transcript levels in 15 day old gemmae of wild type, the four Mp*FRH1*[GOF] mutant lines and four *proACT:FRH1* lines with a strong few rhizoid phenotype. Mp*RSL1* transcript levels were normalised against Mp*APT1* and *MpCUL3*..

DOI: https://doi.org/10.7554/eLife.38529.007

The following figure supplements are available for figure 3:

**Figure supplement 1.** Stem-loop PCR detection of the MpFRH1 miRNA.

DOI: https://doi.org/10.7554/eLife.38529.008

**Figure supplement 2.** Predicted targets of MpFRH1 miRNA.

DOI: https://doi.org/10.7554/eLife.38529.009

**Figure supplement 3.** Transcripts of one of the four predicted MpFRH1 miRNA targets are less abundant in Mp*FRH1*[GOF2] compared to wild type.

DOI: https://doi.org/10.7554/eLife.38529.010

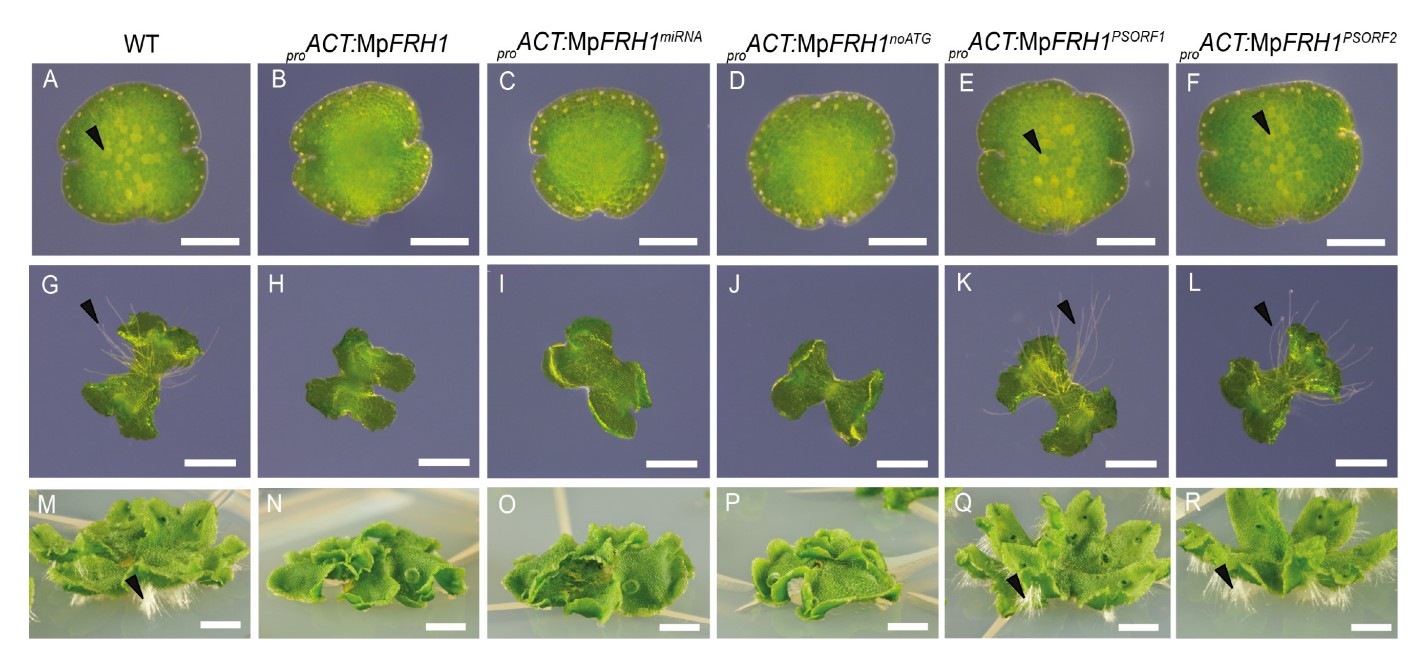

| WT | _pro_ACT:Mp_FRH1_ | _pro_ACT:Mp_FRH1^miRNA_ | _pro_ACT:Mp_FRH1^noATG_ | _pro_ACT:Mp_FRH1^PSORF1_ | _pro_ACT:Mp_FRH1^PSORF2_ |

**Figure 4.** Mp_FRH1_ functions as miRNA. One day old gemma (A-F, scale bar 200 µm), four day old gemma (F-L, scale bar 1 mm) and 28 day old gemma (M-R, scale bar 5 mm) of wild type (**A, G, M**), _pro_ACT:Mp_FRH1_ (**B, H, N**), _pro_ACT:Mp_FRH1^miRNA_ (**C, I, O**), _pro_ACT:Mp_FRH1^noATG_ (**D, J, P**), _pro_ACT: Mp_FRH1^PSORF1_ (**E, K, Q**) and _pro_ACT:Mp_FRH1^PSORF2_ (**F, L, R**). The arrowheads indicate rhizoid precursor cells (in A-F) and rhizoids (in G-R).
DOI: https://doi.org/10.7554/eLife.38529.011

The following figure supplement is available for figure 4:

**Figure supplement 1.** Frequencies of different phenotypes observed in sporelings transformed with partial or modified versions of the Mp_FRH1_ transcript sequence driven by the strong constitutive rice actin promoter.
DOI: https://doi.org/10.7554/eLife.38529.012

mutated to ATC (Mp_FRH1^noATG_), and over-expressed this version behind the strong constitutive rice ACTIN1 promoter in the wild type background. Transformed plants over-expressing _pro_ACT: Mp_FRH1^noATG_ developed very few rhizoids similar to the four Mp_FRH1^GOF_ lines and _pro_ACT:Mp_FRH1_ lines (**Figure 4D,J,P**, **Figure 4—figure supplement 1**). These results are consistent with the hypothesis that Mp_FRH1_ functions as an RNA. To verify that the PSORFs do not encode peptides, we over-expressed wild type versions of two PSORFs that do not overlap with the miRNA encoding fragment in the wild type background. Transformed plants overexpressing either _MpFRH1^PSORF1_ or _MpFRH1^PSORF2_ did not have defects in rhizoid development and were undistinguishable from wild type plants (**Figure 4E–F,K–L,Q–R**, **Figure 4—figure supplement 1**). This is consistent with the hypothesis that the Mp_FRH1_ transcript encodes a miRNA and not a peptide. Together, these findings indicate that Mp_FRH1_ encodes a miRNA that represses rhizoid development.

## MpFRH1 miRNA targets _RSL_ class I gene Mp_RSL1_ mRNA

We identified four putative MpFRH1 miRNA target mRNAs using TargetFinder with default parameters (**Fahlgren and Carrington, 2010**), TargetFinder. GitHub. https://github.com/carringtonlab/TargetFinder) (**Figure 3—figure supplement 2**). One of the predicted targets was a 21 bp sequence on the Mp_RSL1_ mRNA. Therefore, we hypothesised that MpFRH1 miRNA binds directly to Mp_RSL1_ mRNA causing post-transcriptional silencing, either through translational inhibition or mRNA cleavage. To test this hypothesis, we measured the steady state levels of Mp_RSL1_ transcript in Mp_FRH1^GOF_ mutants and _proACT1:MpFRH1_ lines. The steady state levels of Mp_RSL1_ mRNA were lower in all four Mp_FRH1^GOF_ mutants and the four _proACT1:MpFRH1_ lines tested (**Figure 3C**). This is consistent with the hypothesis that MpFRH1 miRNA targets the Mp_RSL1_ mRNA. To test if MpFRH1 suppresses Mp_RSL1_ through mRNA cleavage we performed a 5'RLM-RACE PCR (RNA-ligase mediated rapid amplification of complementary DNA ends PCR) assay. The amplified Mp_RSL1_

mRNA fragment terminated within the predicted MpFRH1 target site (*Figure 3B*, *Supplementary file 3*), indicating MpFRH1 miRNA mediates cleavage of the Mp*RSL1* mRNA. The steady state levels of each of the three other predicted target mRNAs were similar in wild type and Mp*FRH1* gain of function mutants suggesting that they are not MpFRH1 miRNA targets (*Figure 3— figure supplement 3*). Furthermore, the predicted MpFRH1 miRNA target site on the orthologs of each of these three mRNAs is not conserved among liverworts (*Supplementary file 4*). This suggests that none of these three mRNAs is an MpFRH1 miRNA target. We conclude that MpFRH1 miRNA negatively regulates rhizoid development by mediating the cleavage of the Mp*RSL1* mRNA.

## An Mp*FRH1*-resistant form of Mp*RSL1* induces rhizoid development in the Mp*FRH1*$^{GOF2}$ mutant background

If Mp*FRH1* targets the Mp*RSL1* mRNA for cleavage, we predicted that overexpression of an MpFRH1 miRNA-resistant version of Mp*RSL1* would suppress the Mp*FRH1*$^{GOF}$ few rhizoids pheno-type. To test this hypothesis we generated MpFRH1 miRNA-resistant version of Mp*RSL1* (Mp*RSL1*$^{res}$) by introducing seven point mutations in the predicted MpFRH1 miRNA target site on the Mp*RSL1* mRNA (*Figure 5—figure supplement 1*). We then expressed Mp*RSL1*$^{wt}$ and Mp*RSL1*$^{res}$ using the strong Mp*EF1α* promoter in the Mp*FRH1*$^{GOF2}$ mutant background and scored the resulting pheno-types. First, we scored rhizoid production on 26 independent T1 lines of Mp*FRH1*$^{GOF2}$ mutants transformed with the wild type version of Mp*RSL1* ($_{pro}$Mp*EF1α*:Mp*RSL1*$^{wt}$). Twenty-five out of the 26 transformants were rhizoidless like Mp*FRH1*$^{GOF2}$ plants, while only one transformant developed rhi-zoids (*Figure 5—figure supplement 1*). This indicates that expression of the wild type version of Mp*RSL1* from the Mp*EF1∝* promoter does not suppress the rhizoidless phenotype of Mp*FRH1*$^{GOF2}$. We scored rhizoid development in ten lines transformed with the MpFRH1 miRNA resistant version of Mp*RSL1* ($_{pro}$Mp*EF1α*:Mp*RSL1*$^{res}$). Seven of these transformed lines developed abundant rhizoids on the ventral thallus surface and two lines developed rhizoids on both ventral and dorsal surfaces of the thallus, while a single line was rhizoidless and was phenotypically identical to Mp*FRH1*$^{GOF2}$ plants (*Figure 5—figure supplement 1*). These data are consistent with the hypothesis that Mp*RSL1* is a target of the Mp*FRH1* miRNA. To verify that Mp*RSL1* is a target of the Mp*FRH1* miRNA we observed rhizoid formation on gemmae that developed in the transformant lines. We randomly selected three independent T1 lines of each genotype and scored for rhizoid formation on 3 day old gemmalings. None (0%) of the gemmalings expressing the wild type Mp*RSL1* in the Mp*FRH1*$^{GOF2}$ background ($_{pro}$Mp*EF1α*:Mp*RSL1*$^{wt}$ Mp*FRH1*$^{GOF2}$) developed rhizoids (*Figure 5*). In contrast, most (88.9%) gemmalings transformed with the miRNA resistant form of the Mp*RSL1* ($_{pro}$Mp*EF1α*: Mp*RSL1*$^{res}$ Mp*FRH1*$^{GOF2}$) developed rhizoids (*Figure 5*). Gemmae developed in the gemma cups of Mp*FRH1*$^{GOF2}$ plants transformed with $_{pro}$Mp*EF1α*:Mp*RSL1*$^{res}$, while similar to the Mp*FRH1*$^{GOF2}$ plants the gemma cups of $_{pro}$Mp*EF1α*:Mp*RSL1*$^{wt}$ Mp*FRH1*$^{GOF2}$ plants were mostly empty (*Figure 5— figure supplement 1*). The $_{pro}$Mp*EF1α*:Mp*RSL1*$^{res}$ plants with both ventral rhizoids and ectopic dor-sal rhizoids never developed gemmae cups (*Figure 5—figure supplement 1*) and were therefore excluded from this analysis. The restoration of gemma development by the $_{pro}$Mp*EF1α*:Mp*RSL1*$^{res}$ construct in the Mp*FRH1*$^{GOF2}$ background demonstrates that Mp*RSL1* is a target of Mp*FRH1* regula-tion during gemma development. Together these data are consistent with the hypothesis that Mp*FRH1* negatively regulates Mp*RSL1* by mediating the cleavage of the Mp*RSL1* mRNA during the development of structures derived from single epidermal cells in *M. polymorpha*.

## Mp*RSL1* positively regulates Mp*FRH1* transcript level

In biological networks equilibrium is commonly achieved through negative feedback loops, in which positive regulators promote the expression of their repressors. Therefore, we hypothesised Mp*RSL1* may also promote MpFRH1 expression. To test this hypothesis, we measured steady state levels of Mp*FRH1* transcript in the Mp*RSL1* loss-of-function and gain-of-function mutants. Steady state Mp*FRH1* transcript levels were lower than wild type in the Mp*rsl1-1* and Mp*rsl1-2* loss-of-function mutant alleles, but higher than wild type in the Mp*RSL1* gain-of-function mutant alleles Mp*RSL*$^{GOF1-3}$ (*Figure 6*). These results indicate that Mp*RSL1* positively regulates steady state Mp*FRH1* transcript level. However, functional Mp*RSL1* is not required for baseline Mp*FRH1* expression; a low level of Mp*FRH1* mRNA persists in the Mp*rsl1* complete loss-of-function mutant background (*Figure 6*), sug-gesting that other mechanisms also contribute to the regulation of Mp*FRH1* expression. The positive

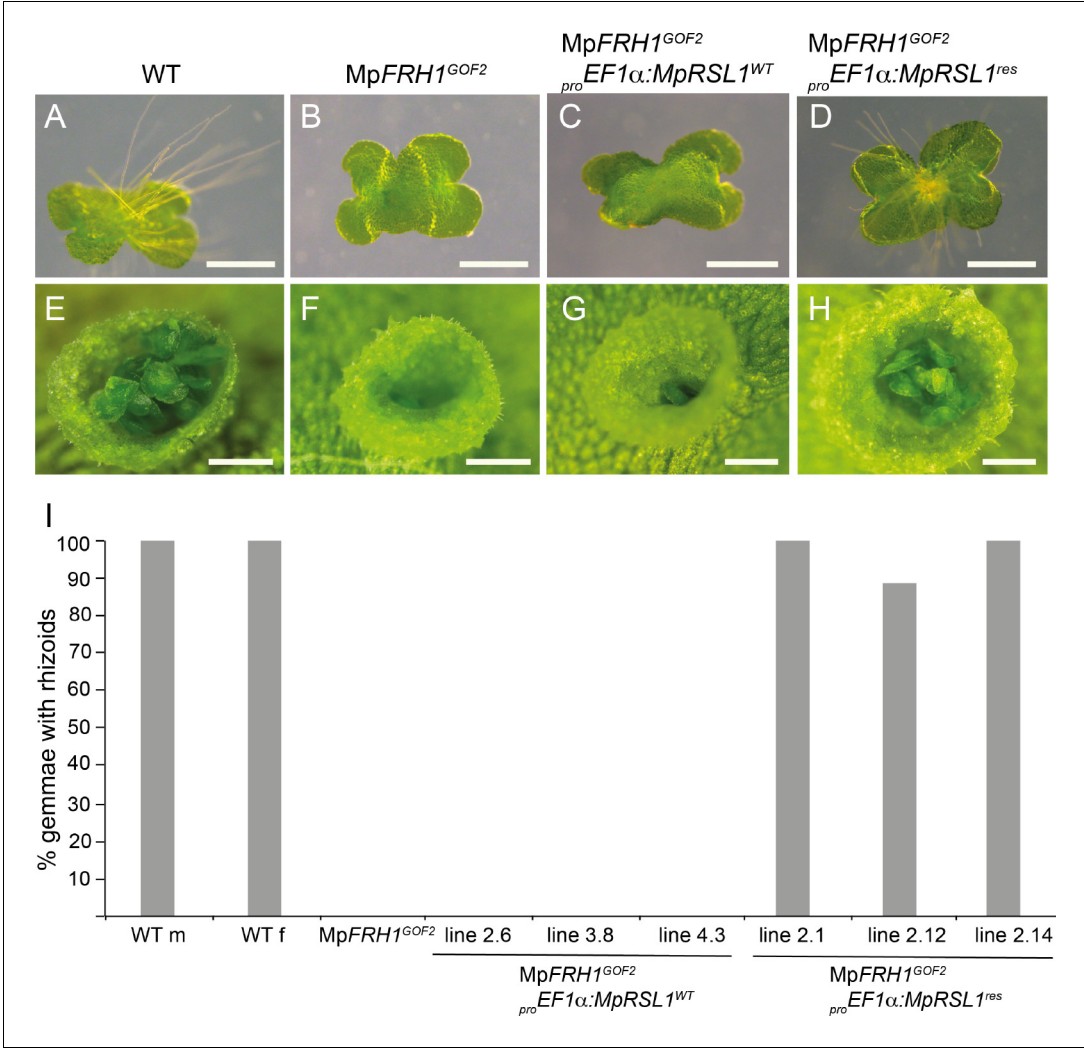

**Figure 5.** MpFRH1 miRNA-resistant form of MpRSL1 restores rhizoid and gemma development in the MpFRH1$^{GOF2}$ mutant background. Three day old gemmae (A–D) of WT (A), MpRSL1$^{GOF2}$ (B), MpFRH1$^{GOF2}$; $_{pro}$EF1a:MpRSL1$^{WT}$ (C) and MpFRH1$^{GOF2}$; $_{pro}$EF1a:MpRSL1$^{res}$ (D), scale bar 500 µm. Gemma cup of mature WT (E), MpRSL1$^{GOF2}$ (F), MpFRH1$^{GOF2}$; $_{pro}$EF1a:MpRSL1$^{WT}$ (G) and and MpFRH1$^{GOF2}$; $_{pro}$EF1a:MpRSL1$^{res}$ (H), scale bar 500 µm. (I) Percentage of three day old gemmae forming rhizoids, n = 18 for each line.
DOI: https://doi.org/10.7554/eLife.38529.013

The following figure supplement is available for figure 5:

**Figure supplement 1.** MpFRH1 miRNA-resistant version of MpRSL1 (MpRSL1$^{res}$) suppresses the MpFRH1$^{GOF2}$ few rhizoids and few gemmae phenotype.
DOI: https://doi.org/10.7554/eLife.38529.014

regulation of MpFRH1 expression by MpRSL1, which is in turn targeted by MpFRH1 miRNA, indicates that MpRSL1 and MpFRH1 miRNA form a regulatory loop with negative feedback.

## The MpFRH1 promoter is expressed in rhizoid precursor cells, rhizoids and epidermal papillae

To identify the cells in which the MpFRH1 promoter is active, we transformed wild type *M. polymorpha* with a reporter gene encoding three copies of the yellow fluorescent protein fused to a nuclear localization signal under the transcriptional control of the 3.5 kb genomic sequence upstream of the MpFRH1 transcript (*proMpFRH1:3xYFP:NLS*). We observed YFP fluorescence, indicative of MpFRH1 promoter activity, in the rhizoid precursor cells, which on young gemmae can be distinguished from

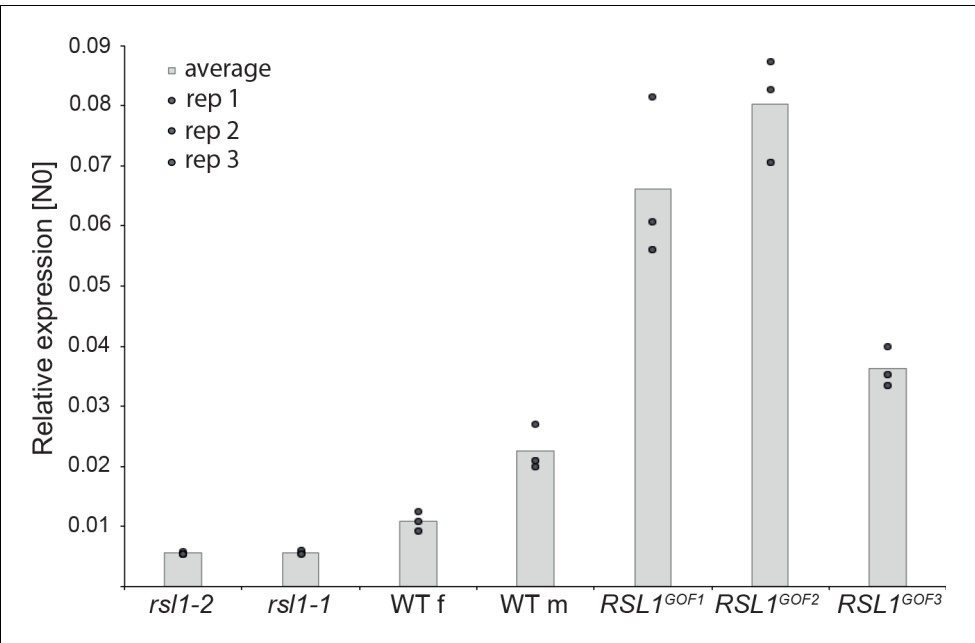

**Figure 6.** Mp*RSL1* positively regulates Mp*FRH1* transcript level. qRT-PCR quantification of steady state Mp*FRH1* transcript levels in 15 day old gemmae of wild type *M. polymorpha*, Mp*rsl1* loss-of-function mutants Mp*rsl1-1* and Mp*rsl1-2*, and Mp*RSL1* gain-of-function mutants Mp*RSL1*<sup>GOF1</sup>, Mp*RSL1*<sup>GOF2</sup> and Mp*RSL1*<sup>GOF3</sup>. Mp*FRH1* transcript levels were normalised against Mp*APT1* and Mp*CUL3*.

DOI: https://doi.org/10.7554/eLife.38529.015

non-rhizoid precursor cells based on their strongly reduced chlorophyll autofluorescence (*Figure 7*). The YFP signal persisted in elongating rhizoids (*Figure 7*). To verify that the YFP signal in the rhizoid precursor cells of *proFRH1:3xYFP-NLS* gemmae is a result of differential expression of the promoter between rhizoid precursor cells and non-rhizoid precursor cells, we analysed as a control the pattern of 3xYFP:NLS expression driven by the ubiquitously expressed 3.5 kb *M. polymorpha INCOMPLETE ROOT HAIR ELONGATION* (Mp*IRE*) promoter (for more details see *Supplementary file 5*). Mp*IRE* promoter expression was detected in both rhizoid precursor cells and non-rhizoid precursor cells (*Figure 7—figure supplement 1*). These results suggest that the strong *proFRH1:3xYFP:NLS* signal detected in rhizoid precursor cells results from stronger Mp*FRH1* promoter activity in these cells than in surrounding cells. In addition, we observed strong Mp*FRH1* promoter activity in mucilage papillae that form near the gemmae meristematic region of 1 day old gemmae (*Figure 7*). We also observed some mucilage papillae without YFP signal, suggesting that Mp*FRH1* is transiently expressed during mucilage papilla development. Taken together, these data indicate that Mp*FRH1* is expressed in epidermal cells that develop rhizoids and papillae.

## Negative regulation of *RSL* class I genes by FRH1 miRNA evolved early in the liverwort lineage

To identify when the regulation of *RSL* class I genes by the MpFRH1 miRNA originated, we searched for the MpFRH1 miRNA target site sequence in *RSL* class I mRNAs among the major land plant lineages. The FRH1 miRNA binding site is 100% conserved in all twelve liverwort species for which *RSL* class I transcript sequence data was available through the 1000 plants project (https://db.cngb.org/onekp/, *Figure 8*, a longer region of the alignment is provided in *Figure 8—figure supplement 1*). These twelve liverwort species belong to Marchantiopsida and Jugermanniopsida; two of the three major liverwort lineages (sequence data for *RSL* class I transcripts in the third major liverwort lineage Haplomitriopsida was not available) (*Forrest et al., 2006*). There is no Mp*FRH1* binding site sequence in the class I *RSL* transcripts of the moss *P. patens,* the lycophyte *S. Kraussiana* or the angiosperm *A. thaliana.* Together, these data suggest that while the regulation of *RSL* class I genes by FRH1-like miRNAs became established early in the liverwort lineage, different negative regulation

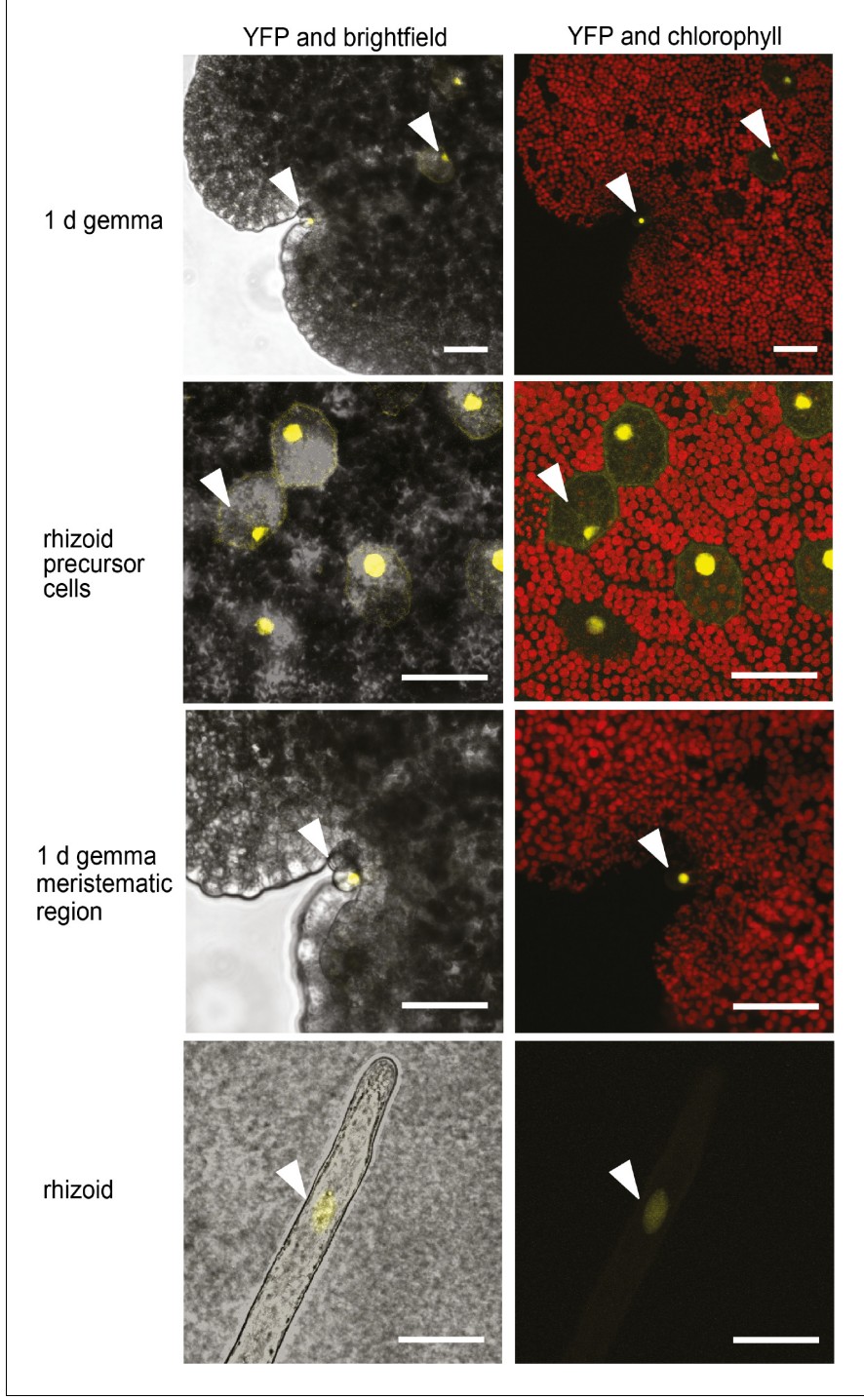

**Figure 7.** Mp*FRH1* is expressed in rhizoid precursor cells, epidermal papillae and rhizoids. Pattern of $_{promoter}$Mp*FRH1:3xYFP-NLS* expression in 1 day old gemmae. The arrowheads indicate rhizoid precursor cells, epidermal papillae and rhizoids. Scale bar 50 µm.

DOI: https://doi.org/10.7554/eLife.38529.016

The following figure supplement is available for figure 7:

**Figure supplement 1.** Mp*FRH1* is expressed in rhizoid precursor cells.

DOI: https://doi.org/10.7554/eLife.38529.017

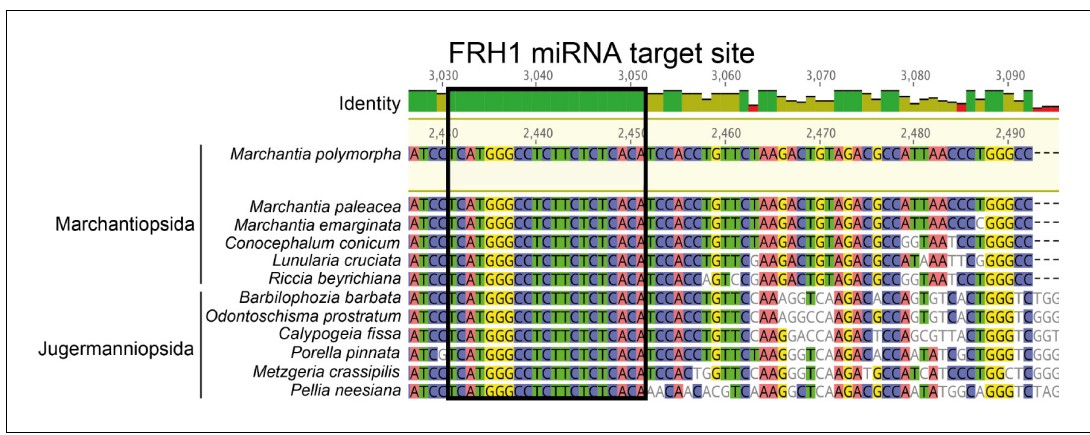

**Figure 8.** The MpFRH1 miRNA target site is conserved in liverwort *RSL* class I transcripts. Liverwort *RSL* class I transcript alignment. The MpFRH1 miRNA target site is circled in black. Longer region of the alignment is provided in *Figure 8—figure supplement 1*.

DOI: https://doi.org/10.7554/eLife.38529.018

The following figure supplement is available for figure 8:

**Figure supplement 1.** The MpFRH1 miRNA target site is conserved in liverwort *RSL* class I transcripts.
DOI: https://doi.org/10.7554/eLife.38529.019

evolved in other lineages. In Arabidopsis, *RSL* class I genes are negatively regulated by the homeo-domain protein GL2. In Arabidopsis GL2 represses *RSL* class I gene expression by directly binding L1 box sequences on the promoters of *RSL* class I genes (*Lin et al., 2015*). The genomic sequence 3.5 kb upstream of Mp*RSL1* does not contain any L1-box sequences. Therefore, the MpFRH1 miRNA negatively regulates Mp*RSL1* in *M. polymorpha* and we find no evidence for a role of class IV home-odomain protein in the negative regulation of Mp*RSL1* expression.

## Discussion

The evolution of form in living organisms results from modulation of gene regulatory networks (also known as GRNs) that comprise relatively ancient conserved elements and more recently evolved elements that control lineage specific traits. Land plant *RSL* class I genes control an ancient gene regu-latory network that positively regulates the development of structures derived from single epidermal cells (*Menand et al., 2007*; *Proust et al., 2016*). The *RSL* class I mechanism is conserved among the major lineages of land plants (*Menand et al., 2007*; *Proust et al., 2016*; *Zalewski et al., 2013*; *Kim et al., 2017*). This suggests that *RSL* class I genes were active in the last common ancestor of the extant land plants, where they controlled the development of structures such as rhizoids that anchored these plants to their substrates. Here we report the discovery of a liverwort-specific miRNA, MpFRH1, that represses the *RSL* class I transcription factor during the development of struc-tures – rhizoids, mucilage papillae and gemmae – that develop from single epidermal cells in *M. pol-ymorpha*. The conservation of the FRH1 target site among liverwort *RSL* class I mRNAs indicates that the FRH1 miRNA likely evolved early in the liverwort lineage or just before the divergence of the liv-erwort lineage from other land plants.

The discovery that a miRNA represses *RSL* class I function in liverworts demonstrates that differ-ent mechanisms of negative regulation of these conserved transcription factors evolved in the liver-worts and angiosperms. While the expression of the two *RSL* class I genes, At*RHD6* and At*RSL1,* positively regulates the development of root hair cells in the root epidermis in the angiosperm *A. thaliana,* these genes are negatively regulated by a homeodomain-leucine-zipper protein AtGL2 (*Di Cristina et al., 1996*; *Bernhardt et al., 2005*; *Bernhardt et al., 2003*; *Koshino-Kimura et al., 2005*) and not by a microRNA. To date, AtGL2 is the only repressor of At*RHD6* and At*RSL1* to have been identified. There are no FRH1 miRNA target sequences in either At*RHD6* or At*RSL1* mRNA and the FRH1 miRNA has not been identified in *A. thaliana* or any other angiosperm. This demon-strates that the FRH1 miRNA does not regulate these genes in *A. thaliana*. The observation that *RSL*

class I genes are repressed by a miRNA and a homeodomain-leucine-zipper transcription factor in *M. polymorpha* and *A. thaliana* respectively demonstrates that at least two independent mechanisms that negatively regulate the ancient *RSL* class I gene mediated differentiation module have evolved among land plants.

It is likely that different modes of negative regulation for *RSL* class I genes have evolved more than twice during the course of land plant evolution. The *RSL* class I genes Pp*RSL1* and Pp*RSL2* promote rhizoid and mucilage papilla development in the moss *P. patens* (*Menand et al., 2007*; *Proust et al., 2016*), but nothing is known about their negative regulation. Similarly, *RSL* class I genes also positively regulate root hair development in the grasses *Oryza sativa* and *Brachypodium distachyon* (*Zalewski et al., 2013*; *Kim et al., 2017*), but the mechanism of their negative regulation is unknown. The FRH1 miRNA target sites are not conserved in these *RSL* class I mRNAs and FRH1 miRNA has not been identified in either mosses or grasses. Therefore, the FRH1 miRNA is unlikely to act as negative regulator outside the liverworts. We conclude that other modes for the negative regulation of *RSL* class I genes evolved among other lineages, but they remain to be discovered.

MpFRH1 miRNA is only found in one monophyletic lineage of land plants, the liverworts. It has been conserved since the divergence of the two major clades of liverworts – Marchantiopsida and Jungermanniopsida – estimated to have occurred more than 405 million years ago (*Morris et al., 2018*). Therefore, we conclude that the FRH1 miRNA may have existed since soon after the divergence of liverworts from other land plant lineages approximately 440 million years ago (*Morris et al., 2018*). This indicates that although the negative regulatory mechanisms have changed in different lineages, FRH1 mediated negative regulation of *RSL* class I genes has been extant since a period in Earth history when the radiation in morphological diversity of land plants occurred. The appearance of the FRH1 miRNA may have been associated with gene regulatory network rewiring that occurred during the morphological diversification early in the liverwort lineage or during the evolution of liverworts from a common, non-liverwort ancestor.

Negative regulators can define when and where positive regulators are expressed and therefore are a key component in any gene regulatory network. For example, many mechanisms for spatial patterning of cell differentiation are based on lateral inhibition by mobile negative regulators. Here a stochastic change in gene expression in a differentiating cell results in the production of mobile negative regulators that suppress differentiation in neighbouring cells. This principle underpins the delta-notch signalling system that defines spacing patterns of different cell types in metazoan tissues and organs (*Collier et al., 1996*). Another example are the CAPRICE family of mobile negative transcriptional regulators that control the pattern of root epidermal cell differentiation in *A. thaliana*. CAPRICE proteins are expressed in non-hair cells, but accumulate in hair-forming cells where they form a protein complex that binds to the promoter of At*GL2* repressing its transcription (*Wada et al., 2002*; *Schellmann et al., 2002*; *Kurata et al., 2005*; *Lee and Schiefelbein, 2002*). CAPRICE-mediated At*GL2* repression facilitates *RSL* class I expression, which then promotes the differentiation of root hair cells (*Lin et al., 2015*; *Wada et al., 2002*). MpFRH1 miRNA produced in cells that go on to develop a rhizoid, mucilage papilla or gemma may negatively regulate *RSL* class I expression in surrounding cells and may have a role in the spatial specification of cell types during patterning the outer surface of the liverwort body. Therefore, repression of rhizoid, mucilage papilla or gemma differentiation could involve the non-cell autonomous repression of Mp*RSL1* expression by mobile MpFRH1 miRNA.

In some cases the expression domain of the negative regulators and their targets are not spatially separated. For example, the final cell size in elongating Arabidopsis root hairs is defined by *RSL* class II gene At*RSL4*, which positively regulates root hair elongation, and two transcription factors AtGTL1 and AtDF1, which negatively regulate root hair elongation by directly repressing the transcription of At*RSL4* and other genes involved in root hair elongation (*Yi et al., 2010*; *Shibata et al., 2018*). Overlapping expression domain have also been observed for miRNAs and their targets. The Arabidopsis miRNA miR164 co-localises with its targets CUP-SHAPED COTYLEDON1 (CUC1) and CUC2 transcription factor mRNAs in the margins of young leaf and floral primordia (*Nikovics et al., 2006*; *Sieber et al., 2007*). miR164 resistant CUC1 and CUC2 maintain the same expression domain as the wild type proteins, but are expressed at a higher level (*Sieber et al., 2007*). These findings suggest that miR164 functions to fine-tune the levels of CUC1 and CUC2 expression within their expression domain (*Sieber et al., 2007*). MpFRH1 may function in similar manner by temporally fine-tuning Mp*RSL1* levels. These hypotheses remains to be tested.

It is possible that evolution of novel negative regulatory mechanisms was involved in the radiation of morphological diversity that followed the colonisation of the land by plants. The morphology of extant streptophyte algae suggests that the streptophyte algal ancestors of land plants had little cell-type diversity and did not develop distinct organs (*McCourt et al., 2004*). However, recent studies demonstrate that many transcription factor families previously thought to have evolved within land plants were already established in streptophyte algae (*Wilhelmsson et al., 2017*; *Hori et al., 2014*). The evolution of morphologically complex land plants from these algal ancestors is likely to have involved the emergence of novel negative regulatory mechanisms – such as miRNAs and transcriptional repressor proteins – around this core set of ancient transcription factors resulting in the evolution of novel gene regulatory networks that programmed novel morphologies. Furthermore, the evolution of new and distinct negative regulatory mechanisms in the different lineages may have underpinned the radiation of morphological diversity in the stem groups of the major lineages of land plants. If correct, it suggests that the radiation in morphological diversity between the Ordovician and Late Devonian resulted, at least in part, from the evolution of novel negative regulatory activities that modulated more ancient and conserved gene regulatory networks that are conserved in many extant land plant lineages. This hypothesis can be tested by defining the mechanism of negative regulation of conserved gene regulatory networks that exist among the main lineages of land plants.

## Materials and methods

### Plant material and growth conditions

*M. polymorpha* accessions Tagaragaike-1 (Tak-1, male) and Tagaragaike-2 (Tak-2, female) (*Ishizaki et al., 2008*) were used as wild type. Plants were grown as described in *Honkanen et al. (2016)*. When plants were grown for RNA extraction the amount of agar on plates was reduced to 1% (w/v) to avoid damaging the rhizoids when detaching plants from the agar.

### Plant transformation

Agrobacterium (GV3101) mediated T-DNA transformation of haploid *M. polymorpha* spores was performed as described in *Honkanen et al. (2016)*. The T-DNA mutant screen, identification of T-DNA flanking genomic sequences and co-segregation analysis were carried out as described in *Honkanen et al. (2016)*.

### Plasmid construction

#### Generation of MpFRH1 pri-miRNA over-expression constructs

*MpFRH1* transcript sequence was amplified from wild type *M. polymorpha* cDNA using Phusion High-Fidelity DNA Polymerase (New England Biolabs) in combination with gene specific primers (TCGGCACTCTCTTCTGTACA, GGCAAAGCAAATTTATTGACGGG). The resulting PCR product was recombined into the pCR8/GW/TOPO Gateway entry vector (Invitrogen). Gateway entry vectors containing the *MpFRH1* transcript variants were synthesised by Life Technologies GeneArt sequence synthesis service. To create over-expression vectors for plant transformation, LR reaction was carried out between the entry vectors and the plasmid proOsACT:Gateway:term-pCam (*Breuninger et al., 2016*).

#### Generation of proMpFRH1::NLS-3xYFP and proMpIRE::NLS-3xYFP

The *MpFRH1* pri-miRNA promoter was analysed using Softberry TSSP promoter prediction (*Solovyev and Shahmuradov, 2003*). *MpFRH1* 3.5 kb promoter fragment including a predicted TATA box and 3.5 kb upstream sequence was amplified from wild type DNA using Phusion High-Fidelity DNA Polymerase (New England Biolabs) with gene specific primers (GAATTCATTTAAA TGAAATCTGAGTTTCC, GGTACCAGGGAGAAAGAGCGCCTGCG). The resulting PCR fragment was cloned between *EcoRI* and *KpnI* restriction enzyme sites on the pCambia 1300 plasmid containing NLS-3xYFP (*Breuninger et al., 2016*). The *proFRH1-3xYFP-NLS* fragment was then amplified using primers TAACAATTTCACACAGGAAAC and AACGACAATCTGATCCAAGCTC, and cloned into pGEM-T Vector (Promega). The *ProFRH1-3xYFP-NLS* fragment was digested out of the pGEM-T Vector using *EcoRI* and ligated into the *EcoRI* site of pCambia1300.

To create a *MpIRE* promoter construct a 3.5 kb fragment upstream of the predicted coding sequence was amplified from wild type DNA using Phusion High-Fidelity DNA Polymerase (New England Biolabs) in combination with gene specific primers (CCTGTCAAACACTGATAGTTAAA-CAAGATCAGGCTCATCAGACG, TGAACGATCGGGGAAATTCGTTTAAACAAAATTGACCG TGCACGGAAC) containing 16 bp extensions complementary to the pCambia1300 plasmid (*Breuninger et al., 2016*). The *MpIRE* promoter fragment was recombined into *PmeI* site of pCambia1300 using In-Fusion HD Cloning Kit (Clontech) following the manufacturer's protocol. A Gateway cassette was ligated behind the *MpIRE* promoter in the resulting plasmid using the Gateway Vector Conversion System (Life Technologies). To create *proIRE::YFP-3xYFP-NLS* construct LR reaction was carried out between the pCambia1300 containing *proMpIRE-GW* and pENTRY3c plasmid containing *NLS-3xYFP* (*Breuninger et al., 2016*).

## Generation of MpFRH1 miRNA resistant MpRSL1

Mp*RSL1* coding sequence was amplified from wild type *M. polymorpha* cDNA and the resulting PCR product recombined into the pCR8/GW/TOPO Gateway entry vector (Invitrogen) as described in *Proust et al. (2016)*. To generate MpFRH1 miRNA resistant version of Mp*RSL1* (Mp*RSL1$^{res}$*) seven point mutations were introduced in the predicted MpFRH miRNA target site by amplifying the cloned Mp*RSL1* coding sequence first in two fragments with Phusion High-Fidelity DNA Polymerase (New England Biolabs) using M13F (−20) primer GTAAAACGACGGCCAGTG and mutated gene specific reverse primer GTGGATGTCAAACTACTGGCCCACGAGGATGAGCGCTTTAGAG for the first fragment, and mutated gene specific forward primer CATCCTCGTGGGCCAGTAGTTTGACA TCCACCTGTTCTAAGACTG and T7 universal primer TAATACGACTCACTATAGGG for the second fragment. Full-length Mp*RSL1$^{res}$* was then constructed by fusing the two fragments in a PCR reaction containing M13F (−20) primer, T7 universal primer and 1:100 dilution of each of the two gel extracted fragments from the first PCR. The resulting Mp*RSL1$^{wt}$* and Mp*RSL1$^{res}$* fragments were then recombined into plasmid *proMpEF1α:Gateway:term-pMpGWB303* (*Ishizaki et al., 2015*).

## RNA extraction, cDNA synthesis and quantitative RT-PCR

Total RNA was extracted from 15 day old wild type and mutant *M. polymorpha* gemmae using the Direct-zol RNA miniprep kit (Zymo Research) following the manufacturer's protocol. Three biological replicate RNA samples were extracted for each line, each replicate consisting of RNA of six gemmae grown on a separate petri dish. The DNAse treatment was performed using the Turbo DNA-free kit (Life Technologies).

One μg of total RNA was used for cDNA synthesis. cDNA synthesis was carried out in a 20 μl reaction using Protoscript II reverse transcriptase (New England Biolabs) and oligo(dT) in the presence of Murin RNase inhibitor (New England Biolabs) according to the manufacturers protocol. Mp*APT1* and Mp*CUL3* were selected as reference genes (*Saint-Marcoux et al., 2015*). qRT-PCT was performed in the Applied Biosystems 7300 Real-Time PCR System (Life Technologies) with SensiMix SYBR Hi-ROX Kit (Bioline) using the following primers: Mp*APT1* F primer CGAAAGCCCAAGAAGC TACC, R primer GTACCCCCGGTTGCAATAAG, Mp*CUL3* F primer AGGATGTGGACAAGGA TAGACG, R primer GTTGATGTGGCAACACCTTG, Mp*FRH1* F primer ACAGCTCGGGGGCTGCAG-CACAAAT, R primer TCAGGATGGCCAGGGGACACTGAAG, and Mp*RSL1* F primer AGATGAGTC TGGGGCAACC, R primer GGATGAGCGCTTTAGAGTGG. Each primer pair was tested to amplify a single product and have amplification efficiency of 1.9–2. Each biological replicate sample was run in three technical replicates. qPCR data was first analysed using LinRegPCR v2012.0 (*Ruijter et al., 2009*). Average $N_0$ value of the three technical replicates were calculated for each biological replicate sample. Relative mRNA expression levels in each biological replicate sample were then determined by normalizing the N0 of each replicate sample separately against each of the two reference genes (Mp*APT1* and Mp*CUL3*), and combining the two normalized values by using the geometric mean.

## Rapid amplification of complementary DNA ends (RACE) PCR

Fragments of the Mp*FRH1* pri-miRNA transcript were identified in *M. polymorpha* gametophyte transcriptome (*Honkanen et al., 2016*). To obtain the full-length *MpFRH1* pri-miRNA transcript RACE-PCR was carried out using 5'/3' RACE Kit (Roche) following the manufacturer's instructions.

## RNA ligase mediated rapid amplification of complementary DNA ends (RLM-RACE) PCR

The Mp*RSL1* transcript cleavage product was identified by carrying out a RLM-RACE PCR as described in *Llave et al. (2011)*. In short, RNA oligonucleotide adaptor (CGACUGGAGCACGAG-GACACUGACAUGGACUGAAGGAGUAGAAA) was first ligated to 5' ends of total RNA extracted from 15 day old wild type gemmae. The RNA was reverse transcribed into cDNA using Protoscript II reverse transcriptase (New England Biolabs) in combination with a gene specific primer GSP-RSL1 (TCGTTGGAAGGCCAATAGTC). PCR was then carried out using primer ASP-F (CGACTGGAGCAC-GAGGACACTGA) that anneals onto the reverse transcribed RNA adaptor and Mp*RSL1* specific primer nested GSP-RSL1 1 (GCCTTTTCAAGCATGGTGAC). The PCR reaction was diluted 1:100. One ul of the diluted PCR product was used as a template for a second nested PCR, which was carried out using primers nested ASP-F (GGACACTGACATGGACTGAAGGAGTA) and nested GSP-RSL1 2 (CTCTGAGGATCGTTCGCACT). The resulting PCR products were gel purified, cloned into pGEM-T vector (Promega) and transformed into E. coli. The miRNA cleavage site was identified by sequencing plasmids extracted from 12 colonies.

## Small RNA enriched RNA extraction and Stem-loop PCR

Small RNA enriched RNA preparations for stem-loop PCR were prepared from 15 day old wild type and mutant *M. polymorpha* gemmae using mirVana miRNA isolation kit (Life Technologies) following the manufacturer's protocol. RNA concentration was estimated using NanoDrop spectrophotometer (Thermo Fisher Scientific, USA). All samples were diluted to 80 ng/µl, and DNAse treated using Turbo DNA-free kit (Life Technologies). To verify the MpFRH1 miRNA, Stem-loop PCR was carried out as described in *Varkonyi-Gasic et al. (2007)*. First a MpFRH1 miRNA specific reverse transcription step was performed using Protoscript II reverse transcriptase (New England Biolabs) in the presence of Murin RNAse inhibitor (New England Biolabs) in a 20 µl reaction containing 320 ng small RNA enriched RNA and 1 µl of 1 µM MpFRH1 specific stem-loop primer (GTCGTATCCAG TGCAGGGTCCGAGGTATTCGCACTGGATACGACACATTG). Subsequent PCR amplification of 2 µl reverse transcribed MpFRH1 miRNA was performed using PCRBIO Ultra Polymerase (PCR Biosystems) with MpFRH1 miRNA specific forward primer (CGGCGTGTGTGAGAAGAGGC) and a universal reverse primer (GTGCAGGGTCCGAGGT). Resulting amplification products were visualised on a 2% agarose gel containing ethidium bromide.

## MiRNA target prediction and analysis of miRNA target site conservation

Targets of the MpFRH1 miRNA in *M. polymorpha* gametophyte transcriptome (*Honkanen et al., 2016*) were predicted using TargetFinder v1.7 with default parameters (*Fahlgren and Carrington, 2010*, TargetFinder. GitHub https://github.com/carringtonlab/TargetFinder). Mapoly gene ID (*Bowman et al., 2017*) for each transcript was identified using the MarpolBase BLAST server (http://marchantia.info/blast/). Liverwort orthologs of each predicted target were then retrieved from the 1KP database using the protein sequence as a query using the TBLASTN algorithm (https://db.cngb.org/onekp/). Sequence alignment between the predicted targets was carried out using L-INS-I method in MAFFT version 7 (*Katoh and Standley, 2013*). The resulting sequence alignments were visualised using Geneious 9.1.6 (*Kearse et al., 2012*) and BioEdit 7.2.5 (Ibis Biosciences, USA).

## Microscopy and image analysis

For each experiment at least 15 gemmae were observed for each line. Plants were imaged using a Leica DFC310 FX camera connected to a Leica M165 FC stereomicroscope. Confocal laser scanning microscopy was carried out with the Zeiss LSM510 Meta microscope using the Zeiss Plan-Neofluar 25x/0.8 water immersion lens with Argon/2 laser excitation at 488 nm in order to observe fluorescence emitted by YFP and chlorophyll at 505–550 nm and 645–710 nm, respectively. Fluorescence images were constructed by making maximum intensity projections from a Z-stack containing the epidermal cell layer of gemmae. For scanning electron microscopy (SEM) gemmae collected from gemma cups were immediately fixed in dry methanol, critical point dried using a Tousimis Autosam-dri-815, mounted on aluminium stubs and coated with a gold/palladium mixture using a Quorum technologies SC7640 sputter coater. The samples were then imaged with a JEOL JSM-5510 SEM.

All processing of confocal microscopy images was carried out using Fiji (*Schindelin et al., 2012*). Other images were adjusted using Adobe Photoshop CS4.

## Acknowledgements

SH was funded by a European Research Council advanced award (EVO-500) and Biotechnological and Biological Research Council Doctoral Training Award (BB/F016093/1); AT was supported by a Biotechnological and Biological Research Council Doctoral Training Partnership award (J0144271/1) and an EPA Cephalosporin Scholarship. MAAV was funded by UC MEXUS-19941–44 CONACYT-158550 and the Royal Society Newton Advance Fellowship (NA150181) project number RG79985. LD was funded by a European Research Council advanced award (EVO-500) project number 25028. We are grateful to John Baker (Oxford University) for photographic assistance.

## Additional information

### Funding

| Funder | Grant reference number | Author |
| --- | --- | --- |
| European Commission | EVO-500 250284 | Suvi Honkanen<br>Liam Dolan |
| Biotechnology and Biological Sciences Research Council | BB/F016093/1 | Suvi Honkanen<br>Anna Thamm |
| University of Oxford | EPA Cephalosporin Scholarship | Anna Thamm |
| University of California Institute for Mexico and the United States | UCMEXUS-19941-44-OAC7 | Mario A Arteaga-Vazquez |
| Royal Society | Newton Advanced Fellowship NA150181 RG79985 | Mario A Arteaga-Vazquez |
| Biotechnology and Biological Sciences Research Council | J0144271/1 | Suvi Honkanen<br>Anna Thamm |

The funders had no role in study design, data collection and interpretation, or the decision to submit the work for publication.

### Author contributions

Suvi Honkanen, Conceptualization, Data curation, Formal analysis, Validation, Investigation, Visualization, Methodology, Writing—original draft, Writing—review and editing; Anna Thamm, Formal analysis, Validation, Investigation, Visualization; Mario A Arteaga-Vazquez, Formal analysis, Formal analysis of data, Evaluation of data, Input to drafting manuscript, Approval of the final version of the manuscript; Liam Dolan, Conceptualization, Supervision, Funding acquisition, Methodology, Writing—original draft, Writing—review and editing

### Author ORCIDs

Suvi Honkanen (iD) http://orcid.org/0000-0003-3923-3365
Liam Dolan (iD) http://orcid.org/0000-0003-1206-7096

### Decision letter and Author response
Decision letter https://doi.org/10.7554/eLife.38529.031
Author response https://doi.org/10.7554/eLife.38529.032

## Additional files

### Supplementary files
• Supplementary file 1. TAIL-PCR band sequences

DOI: https://doi.org/10.7554/eLife.38529.020

• Supplementary file 2. Mp*FRH1* transcript sequence

DOI: https://doi.org/10.7554/eLife.38529.021

• Supplementary file 3. 5'RLM-RACE PCR band sequences

DOI: https://doi.org/10.7554/eLife.38529.022

• Supplementary file 4. Multiple sequence alignments of the predicted MpFRH1 miRNA target mRNAs in M. polymorpha and their orthologs from other liverworts. The predicted MpFRH1 target site is indicated with a grey arrow. Region around the predicted miRNA target site (top) and overview of the alignment (bottom). (A) Foie gras domain containing protein Mapoly0075s0041.1. (B) Basic helix-loop-helix transcription factor MpRSL1 Mapoly0039s003 (C) Basic helix-loop-helix transcription factor transcript 27676. (D) Nucleotide-rhamnose synthase/epimerase-reductase Mapoly0005s0120.

DOI: https://doi.org/10.7554/eLife.38529.023

• Supplementary file 5. Promoter analysis

DOI: https://doi.org/10.7554/eLife.38529.024

• Transparent reporting form

DOI: https://doi.org/10.7554/eLife.38529.025

## Data availability

All data generated or analysed during this study are included in the manuscript and supporting files.

The following previously published datasets were used:

| Author(s) | Year | Dataset title | Dataset URL | Database, license, and accessibility information |
|---|---|---|---|---|
| Champion C, Hetherington AJ, Kelly S, Saint-Marcoux D, Morieri G, Proust H, Dolan L | 2016 | Marchantia polymorpha subsp. ruderalis, whole genome shotgun sequencing project | https://www.ncbi.nlm.nih.gov/nuccore/LVLJ00000000.1 | Publicly available at the NCBI GenBank (accession no. LVLJ00000000.1) |
| Hetherington AJ, Kelly S, Morieri G, Proust H, Dolan L | 2016 | TSA: Marchantia polymorpha, transcriptome shotgun assembly | https://www.ncbi.nlm.nih.gov/nuccore/1032266567 | Publicly available at the NCBI GenBank (accession no. GEFO00000000.1) |

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
