## [Decision Letter]

Thank you for submitting your article "Negative regulation of conserved *RSL* class I bHLH transcription factors evolved independently among land plants" for consideration by *eLife*. Your article has been reviewed by two peer reviewers, and the evaluation has been overseen by a Reviewing Editor and Christian Hardtke as the Senior Editor. The following individuals involved in review of your submission have agreed to reveal their identity: Michitaro Shibata (Reviewer #1); Shohei Yamaoka (Reviewer #2).

The reviewers have discussed the reviews with one another and the Reviewing Editor has drafted this decision to help you prepare a revised submission.

Summary:

The work begins to illuminate the complexity of molecular mechanisms that mediate development in mosses and how this compares/contrasts to vascular plants.

Essential revisions:

The reviewers have largely provided points of clarification that will help to make the manuscript more approachable.

*Reviewer #1:*

In this manuscript, the authors isolated FEW RHIZOIDS1 (MpFRH1) which negatively regulates rhizoid development by forward genetic screening in *Marchantia polymorpha*. They showed MpFRH1 encodes a novel microRNA targeting MpRSL1 mRNA by genetic approach such as overexpression of FRH1 and miRNA-resistant version of Mp*RSL1*. Furthermore, the authors found RSL1 positively regulates MpFRH1 expression, thus forming a feedback loop.

Although a lot of evidence suggest the importance of *RSL* subfamily in the filamentous root cell development among land plants, how their expression is regulated is not well characterized. This paper clearly showed the mechanism of negative regulation in *Marchantia polymorpha*. It is also interesting that this negative regulation of *RSL* class I gene is conserved only in liverwort, suggesting that different from *RSL* class I gene, "independent negative regulatory mechanisms evolved in different lineages during land plant evolution" as the authors pointed out.

This work is very interesting and the quality of data is generally very high. I have only a few suggestions and comments to improve the quality of this work.

The authors mentioned "There are three putative small open reading frames". However, only two of three were tested. If there are reasons not to test all three potential ORFs, the authors should mention about it.

The authors wrote "To test if MpFRH1 suppresses Mp*RSL1* mRNA cleavage we performed 5'RLM-RACE PCR assay". However, only the model is shown in Figure 3B. The authors should show the data indicating that the cleavage of Mp*RSL1* mRNA is mediated by MpFRH1 miRNA.

The subsection "Mp*RSL1* positively regulates Mp*FRH1* transcript level" starts without any explanation to study the hypothesis. I recommend to mention the motivation for why authors want to test if Mp*RSL1* regulates Mp*FRH1* expression.

For Figure 3—figure supplement 3, do the numbers show the three replicate of experiments in the panel A? If so, the authors need to mention about the weak signal of 3. in Basic helix-loop-helix transcription factor (Mapoly0100s0033).

To prove that MpFRH1 encodes miRNA, I think stem-loop PCR is not sufficient. Northern blot analysis is a more direct and reliable approach to detect miRNA of MpFRH1. If possible, I suggest to include these data to strengthen this point.

*Reviewer #2:*

The authors identified a miRNA named MpFRH1, which negatively regulates rhizoid development in Marchantia. MpFRH1 also regulates development of apical papillae and gemmae, suggesting its role on various types of epidermal cell differentiation. Using multiple mutant versions of MpFRH1, the authors clearly show that MpFRH1 targets Mp*RSL1* transcripts for down-regulation to inhibit rhizoid development. They further show that MpFRH1 target sites are conserved specifically in a part of liverwort species, suggesting divergence of negative regulatory mechanism of RSL1-like transcription factors during land plant evolution. The strategy taken in this study is straightforward and the quality of the evidence and presentation are high. I briefly suggest the following revisions:

The main text states that the Mp*RSL1* mRNA fragment terminated at the putative cleavage site, although Figure 3B only shows the gene structure and the predicted site. The authors should provide such experimental data, or revise the main text and Materials and methods.

The authors clearly show antagonistic relationship between MpFRH1 and Mp*RSL1* during rhizoid development. Overexpression of wild-type Mp*RSL1* by a strong EF1 α promoter did not overcome the MpFRH1 GOF mutation, implying strong degeneration activity of MpFRH1 against Mp*RSL1* transcripts. This raises the question how Mp*RSL1* is activated to specify gemma epidermal cells as the rhizoid precursor cells, in which MpFRH1 are also expressed, as shown by the MpFRH1 promoter activity. It is possible that MpFRH1 expression is significantly reduced, or diminished, before the rhizoid precursor cells differentiate on the gemma epidermal cell layer. More information about spatio-temporal pattern of the MpFRH1 promoter activity during gemma development would be helpful for better understanding of the molecular mechanism of rhizoid precursor cell differentiation. I suppose this could be done by simply observing immature gemmae of the MpFRH1 *promoter:3xYFP-NLS* lines.

---

## [Author Response]

Essential revisions:The reviewers have largely provided points of clarification that will help to make the manuscript more approachable.Reviewer #1:[…] The authors mentioned "There are three putative small open reading frames". However, only two of three were tested. If there are reasons not to test all three potential ORFs, the authors should mention about it.

We did test all the three putative small ORFs individually by over-expressing each in the wild type background (Figure 4—figure supplement 1). One of the small ORFs, which we designated Mp*FRH1^miRNA^*, contains the entire miRNA producing hairpin and therefore over-expression of this 150 bp fragment reproduces the few rhizoids phenotype (Figure 4—figure supplement 1). The other two fragments were designated Mp*FRH1^PSORF1^* and Mp*FRH1^PSORF2^*. The over-expression of the other two putative small ORFs (Mp*FRH1^PSORF1^* and Mp*FRH1^PSORF2^*) did not reproduce the few rhizoids phenotype. We reported this in the text:

“To verify that the PSORFs do not encode peptides, we over-expressed wild type versions of two PSORFs that do not overlap with the miRNA encoding fragment in the wild type background. Transformed plants overexpressing either Mp*FRH1^PSORF1^* or Mp*FRH1^PSORF2^* did not have defects in rhizoid development and were undistinguishable from wild type plants”

The authors wrote "To test if MpFRH1 suppresses MpRSL1 mRNA cleavage we performed 5'RLM-RACE PCR assay". However, only the model is shown in Figure 3B. The authors should show the data indicating that the cleavage of MpRSL1 mRNA is mediated by MpFRH1 miRNA.

The experimental data for 5’RLM RACE PCR has now been included as Supplementary file 3. This is referred to in the subsection "Mp*FRH1* miRNA targets *RSL* class I gene Mp*RSL1* mRNA”.

The subsection "MpRSL1 positively regulates MpFRH1 transcript level" starts without any explanation to study the hypothesis. I recommend to mention the motivation for why authors want to test if MpRSL1 regulates MpFRH1 expression.

We now include a rationale. “In biological networks equilibrium is commonly achieved through negative feedback loops, in which positive regulators promote the expression of their repressors. Therefore, we hypothesised Mp*RSL1* may also promote Mp*FRH1* expression. To test this hypothesis, we measured steady state levels of Mp*FRH1* transcript in the Mp*RSL1* loss-of-function and gain-of-function mutants”

For Figure 3—figure supplement 3, do the numbers show the three replicate of experiments in the panel A? If so, the authors need to mention about the weak signal of 3. in Basic helix-loop-helix transcription factor (Mapoly0100s0033).

Yes, the three replicate experiments are shown in panel A. Two out of three samples were similar to Mp*FRH1^GOF^* and wild type. Therefore we think Mapoly0100s0033 is an unlikely target. With the third replicate sample RNA quantification could be off (Mapoly0075s0041 also gives slightly lower amplification in this replicate), there could be a pipetting error or there could be different levels of contamination from genomic DNA between the replicates (please see below). The target site is also fairly close to the 5’ UTR of the gene and therefore differences in RNA integrity could explain the difference in signal intensities.

Note that the computationally predicted MpFRH1 miRNA target site is not present on the Mapoly0100s0033 transcript found in the Bowman et al., 2017 transcriptome (see the alignment in Supplementary file 4 C). We initially carried out the target prediction using the *M. polymorpha* gametophyte transcriptome published in Honkanen et al., 2016. The predicted Mp*FRH1* miRNA target site is on a short transcript 27676, which is identical to Mapoly0100s0033 transcript in its 5’ region. However the 3’ region of this transcript including the predicted miRNA target site is within the first intron of Mapoly0100s0033. Therefore, the 3’ region of transcript 27676 may be a partial alternatively spliced transcript of Mapoly0100s0033 or the reads may have originated from genomic contamination. The *Marchantia* genome browser (marchantia.info) shows very low level of reads on this region (scaffold_100:358969..360369). Therefore we do not believe the reduction seen in one of the three replicate samples is biologically significant.

To prove that MpFRH1 encodes miRNA, I think stem-loop PCR is not sufficient. Northern blot analysis is a more direct and reliable approach to detect miRNA of MpFRH1. If possible, I suggest to include these data to strengthen this point.

We do not have the Northern blot data. The miRNA cataloguing paper by Tsuzuki et al. 2016 showed that Mp*FRH1* miRNA is expressed and accumulates in *M. polymorpha*. Tsuzuki et al. identified the Mp*FRH1* miRNA (mpo-MIR11861) in each of the three replicate sequencing experiments (Supplementary file 1). This demonstrates that the Mp*FRH1* miRNA exists in *Marchantia polymorpha*. Therefore, Northern blots while useful, are not necessary to demonstrate the presence of the Mp*FRH1* miRNA in *Marchantia polymorpha*.

Reviewer #2:[…] The main text states that the MpRSL1 mRNA fragment terminated at the putative cleavage site, although Figure 3B only shows the gene structure and the predicted site. The authors should provide such experimental data, or revise the main text and Materials and methods.

The experimental data for 5’RLM RACE PCR data has now been included as Supplementary file 3.

The authors clearly show antagonistic relationship between MpFRH1 and MpRSL1 during rhizoid development. Overexpression of wild-type MpRSL1 by a strong EF1 α promoter did not overcome the MpFRH1 GOF mutation, implying strong degeneration activity of MpFRH1 against MpRSL1 transcripts. This raises the question how MpRSL1 is activated to specify gemma epidermal cells as the rhizoid precursor cells, in which MpFRH1 are also expressed, as shown by the MpFRH1 promoter activity. It is possible that MpFRH1 expression is significantly reduced, or diminished, before the rhizoid precursor cells differentiate on the gemma epidermal cell layer. More information about spatio-temporal pattern of the MpFRH1 promoter activity during gemma development would be helpful for better understanding of the molecular mechanism of rhizoid precursor cell differentiation. I suppose this could be done by simply observing immature gemmae of the MpFRH1 promoter:3xYFP-NLS lines.

Rhizoid precursor cells become established very early in gemmae development when it is difficult to image their development. We have not been able to acquire these data yet. We are planning to continue to do this but it will clearly take some time.